# Temporal Evolution and Prognostic Role of Indeterminate Response Sub-Groups in Patients with Differentiated Thyroid Cancer after Initial Therapy with Radioiodine

**DOI:** 10.3390/cancers15041270

**Published:** 2023-02-16

**Authors:** Domenico Albano, Pietro Bellini, Francesco Dondi, Anna Calabrò, Claudio Casella, Stefano Taboni, Davide Lombardi, Giorgio Treglia, Francesco Bertagna

**Affiliations:** 1Nuclear Medicine, University of Brescia, ASST Spedali Civili Brescia, 25123 Brescia, Italy; 2Department of Molecular and Translation Medicine, Surgical Clinic, University of Brescia, 25121 Brescia, Italy; 3Section of Otorhinolaryngology-Head and Neck Surgery, Department of Neurosciences, Azienda Ospedale-Università di Padova, 35131 Padova, Italy; 4Unit of Otorhinolaryngology-Head and Neck Surgery, ASST Spedali Civili di Brescia, Department of Medical and Surgical Specialties, Radiologic Sciences and Public Health, University of Brescia, Piazzale Spedali Civili 1, 25123 Brescia, Italy; 5Nuclear Medicine, Imaging Institute of Southern Switzerland, Ente Ospedaliero Cantonale, 6500 Bellinzona, Switzerland; 6Department of Nuclear Medicine and Molecular Imaging, Lausanne University Hospital, University of Lausanne, 1011 Lausanne, Switzerland; 7Faculty of Biomedical Sciences, Università della Svizzera Italiana, 6900 Lugano, Switzerland

**Keywords:** indeterminate response, thyroid cancer, thyroid, radioiodine, nuclear medicine, therapy

## Abstract

**Simple Summary:**

The main aim of this retrospective study was to investigate the dynamic evolution and prognostic role of patients affected by DTC and IR after initial therapy. From January 2010 to December 2017, 2176 patients who received radioiodine (RAI) for DTC after total or near total thyroidectomy were included. Among them, 288 had IR one year after therapy (187 TgAb+, 76 Tg+, 25 imaging+). IRTg+ patients had a higher probability of evolving into an incomplete response. Only stimulated Tg after one year and nodal disease at diagnosis may predict the final status of the disease. Progression-free survival was significantly shorter in IRTg+ than IRTgAb+ and IRimaging+ groups and in patients with sTg > 3.3 ng/mL.

**Abstract:**

The clinical outcome of patients affected by Differentiated Thyroid Carcinoma (DTC) and an indeterminate response (IR) after initial therapy is not yet clear. IR includes three different sub-groups of patients: (1) IRTg+ group: Detectable thyroglobulin (Tg), regardless of antithyroglobulin antibodies (TgAb) presence or imaging studies; (2) IRTgAb+ group: Positive TgAb, regardless of Tg levels and nonspecific imaging findings; (3) IRImaging+ group: Nonspecific findings on neck ultrasonography or faint uptake in the thyroid bed on the whole-body scan, negative TgAb, and undetectable Tg. The main aim of this retrospective study was to investigate the dynamic evolution and prognostic role of these patients. From January 2010 to December 2017, 2176 patients who received radioiodine for DTC after total thyroidectomy were included. Two-hundred-eighty-eight patients had IR one year after therapy (187 TgAb+, 76 Tg+, 25 imaging+). After two years, 110 patients (38%) were reclassified as an excellent response and 5 (2%) as an incomplete response; after five years, 221 (77%) achieved an excellent response and 11 (4%) showed an incomplete response. One-year stimulated Tg and nodal disease at diagnosis may predict the final status of the disease. Progression-free survival was significantly shorter in IRTg+ than in IRTgAb+ and IRimaging+ groups. Considering Tg+ patients, a threshold of 3.3 ng/mL is best to predict prognosis.

## 1. Introduction

Differentiated thyroid carcinoma (DTC) is a slow-growing tumor with very low disease-specific mortality rates, with the exception of distant metastatic disease, which is associated with a significantly worse prognosis [1,2]. Standard-of-care management for DTC includes risk-adapted surgery, and postoperative radioiodine (RAI) therapy if indicated. However, a dynamic risk-stratification scheme is suggested in the recent American Thyroid Association (ATA) guidelines for individualized risk assessment and management [3]. In the guidelines in Table 13, a new category of response after initial therapy called an indeterminate response (IR) was introduced, which included patients with biochemical and structural findings not suggestive of an excellent or incomplete response. This category had a moderate risk (15–20%) of developing a structural disease during the follow-up and needed personalized management characterized by careful observation with serial and appropriate imaging and serum Thyroglobulin (Tg) and Thyroglobulin Antibodies (TgAb) measurements. This new treatment response class was derived from several studies [4,5] that recommended a separate category for these cases and personalized management, with careful observation and further investigations only in selected cases. According to ATA guidelines [3], IR includes three different subgroups of patients according to imaging and biochemical findings: (1) IRTg+ group: Patients with detectable Tg values (nonstimulated Tg values detectable but less than 1 ng/mL or TSH-stimulated Tg between 1 and 10 ng/mL), regardless of TgAb or imaging findings; (2) IRTgAb+ group: Patients with positive stable or declining TgAb, regardless of Tg levels; (3) IRImaging+ group: Nonspecific findings on neck ultrasonography or faint uptake in the thyroid bed on an RAI whole-body scan, negative TgAb, and undetectable Tg. IRImaging+ group definition is quite vague and operator-dependent, including, for example, patients with avascular thyroid bed nodules with a size less than 1 cm or atypical cervical lymph nodes not biopsied, or weak uptake in the thyroid bed in the RAI whole body scan. However, the possibility to have faint RAI uptake in the thyroid bed after initial therapy (surgery plus RAI) is not negligible and in most cases without unfavorable meaning [6]. The aim of this study was to analyze the dynamic evolution and clinical outcome of patients with DTC and IR after initial therapy focusing on the three different IR subgroups.

## 2. Materials and Methods

We retrospectively screened 2176 patients who underwent RAI therapy for DTC after total or near total thyroidectomy between January 2010 and December 2017, who were hospitalized in our Nuclear Medicine Department for the ablation of thyroid remnant tissue. Among them, a total of 288 (235 females; 53 males; sex ratio F:M 4:1; average age 49 years) patients were classified as having IR after initial therapy according to the ATA guidelines [3]. All patients had histopathological confirmation of DTC: 118 classic variants of papillary carcinoma, 53 follicular carcinoma, 92 follicular variants of papillary carcinoma, and 25 aggressive papillary variants (12 tall-cell variants of papillary carcinoma, 5 columnar cell variants, 2 hobnail variants, and 6 poorly differentiated carcinoma). Tumor size was 16 ± 13 mm (range 1–130 mm); multicentricity of the disease was registered in 168 cases (58%). Patients with unifocal microcarcinoma were excluded because they did not receive RAI. Patients underwent total thyroidectomy with/without central neck or lateral neck dissection and received RAI according to the risk class based on the TNM staging of the American Joint Committee on Cancer/International Union against Cancer currently in use [7] and the risk of structural disease recurrences as suggested by ATA guidelines [3]. Typically, 1.1 GBq was administered to low-risk DTC and 1.85–3.7 GBq to intermediate-risk DTC. In patients without antithyroglobulin antibody (AbTg) interference, serum thyroglobulin (Tg) at the time of the first RAI treatment was 3.4 ± 5.5 ng/mL (range 0.2–114); TgAb was positive in 198 patients (69%) at ablation. The baseline characteristics of all patients are summarized in Table 1. The average follow-up time was 84 ± 16 months (range 60–132 months).

### 2.1. Follow-Up after Initial Therapy

After initial therapy, all patients were started on levothyroxine for thyrotropin suppression and were regularly followed up through the evaluation of consecutive measurements of basal Tg and neck ultrasound performed at 6–12 months intervals. One year after the initial therapy, for the evaluation of the treatment response, all patients underwent diagnostic RAI whole-body scintigraphy (WBS) with the measurement of stimulated Tg and TgAb. All patients stopped levothyroxine 40 days prior to WBS and were put on Levo-triiodothyronine in the first 20 days. The WBS was performed in continuous mode with a high-energy general-purpose (HEGP) parallel hole collimator, 364-keV photopeak with ±10% energy windows settings, and scatter correction. The infrared-based real-time automatic body contouring system was activated for simultaneous dual-view (anterior/posterior) scans with a matrix of 256 × 1024. WBS was followed by a single photon emission computed tomography/computed tomography (SPECT/CT) scan on the same day if equivocal findings were present. SPECT/CT (Infinia Hawkeye II, GE Healthcare, Haifa, Israel) was equipped with 1-in. StarBrite™ Crystal. SPECT images were acquired with the HEGP collimator, with a matrix size of 128 × 128, 364-keV photopeak with ±10% energy and scatter windows, dual-detector 180° acquisition, an angular step of 3°, and timesime per step/view. The CT parameters were 140 kV, 2.5 mA, a rotation speed of 30, 10-mm slice thickness, and a 256 × 256 matrix. CT acquisition was performed with a 2-slice helicoidal acquisition. An ordered subset expectation maximization iterative reconstruction with CT-based attenuation correction and scatter correction was performed. After initial therapy, an indeterminate response was defined as follows: (1) Basal thyroglobulin (bTg) ≥ 0.2 to <1 ng/mL or stimulated Tg (sTg) ≥ 1 to <10 ng/mL; (2) positive TgAb but with stable or declining levels during the duration; or (3) nonspecific findings on neck US or faint uptake in the thyroid bed on diagnostic WBS. In the case of suspected structural or biochemical relapse, further evaluations with additional imaging examinations, such as computed tomography (CT), magnetic resonance imaging, positron emission tomography, or biopsy were performed according to the suggestion of the attending physician, based on the risk of the individual patient. Each patient’s disease status was reassessed two and five years after initial therapy by performing imaging studies (ultrasound, WBS, and blood samples) and was categorized as (1) excellent response, in the case of resolution of non-specific laboratory and imaging tests; (2) IR, in the case of persistence of non-specific biochemical/structural findings; (3) biochemically or structurally incomplete response, in the case of increasing Tg levels above 1 ng/mL on levothyroxine therapy or above 10 ng/mL after Levo-thyroxine withdrawal, or evidence of local or distant disease with any Tg level. In some cases, patients performed WBS and sTg at 2 years, preferring not to be stimulated with Tg at this time; while 5 years after the initial treatment, all patients performed WBS and sTg as part of our institutional protocol. Neck ultrasound and/or other imaging studies according to patient features were performed in years 2 and 5.

### 2.2. Statistical Analysis

All statistical analyses were carried out using the Statistical Package for Social Science (SPSS) version 23.0 for Windows (IBM, Chicago, IL, USA) and MedCalc Software version 17.1 (Ostend, Belgium). Numerical parameters were described as the average, minimum, maximum, and standard deviation (SD), while categorical parameters as absolute and relative frequencies. We applied receiver operating characteristic (ROC) curve analysis to identify the optimal cutoff point of Tg, which interprets the results of progression-free survival (PFS). PFS was calculated from the date of diagnosis of DTC to the date of first disease progression, relapse, death, or the date of the last follow-up. Survival curves were plotted according to the Kaplan–Meier method and differences between groups were analyzed using a two-tailed log-rank test. Cox regression was used to estimate the hazard ratio (HR) and its confidence interval (CI). A *p*-value of <0.05 was considered statistically significant.

## 3. Results

### 3.1. Natural Course of IR Disease

Two years after therapy, 110 (38%) patients spontaneously achieved an excellent response; 173 (60%) remained in the IR group, while 5 had progressive disease and showed an incomplete response (two biochemically incomplete responses and three structurally incomplete responses). However, after 5 years, an excellent response was present in 221 (77%) cases, IR in 56 (19%), and an incomplete response in the remaining 11 cases (4%), with 8 structurally and 3 biochemically incomplete responses (Figure 1a).

RAI activity at ablation was not related to the risk of developing IR 1 year after the first therapy (2.85 GBq vs. 2.6 GBq in the whole population, *p* = 0.345).

The RAI activities injected at ablation were not related to the class response categories. Among patients who received 1.1 GBq (*n* = 900), 117 (13%) had IR; while among the patients who received more than 1.1 GBq (*n* = 1276), 171 (13%) developed IR.

Among 288 patients with IR one year after the initial therapy, 187 (65%) belonged to the IRTgAb+ group, 76 (26%) to the IRTg+ group, and 25 (9%) to the IRimaging+ group. Of the 173 patients with IR two years after initial treatment, 136 (79%) were IRTgAb+, 31 (18%) were IR Tg+, and 6 (3%) were IRImaging+. Instead, after five years, 52 (90%) patients had IR due to TgAb+, 4 (7%) due to Tg+, and 2 (3%) due to Imaging+ (Figure 1b). Fifty-four patients received a new RAI treatment and belonged to the following groups: 24 with IRTg+, 27 with IRTgAb+, and 3 with IRImaging+. The decision to perform a new therapy was made by the nuclear medicine physician according to imaging and biochemical evidence. Twenty-four patients underwent new therapy due to the presence of positive sTg (often near 10 ng/mL, always high, then 5 ng/mL); 27 for the presence of positive TgAb especially when the trend of Tgab was stable and concomitant with non-negative WBS; and 3 for the presence of markedly positive WBS, such as big thyroid remnants.

In the IRTg+ group (*n* = 76), two years after therapy, 46 patients reached an excellent response, 26 showed IR, and 4 developed an incomplete response, while in the fifth year after treatment, out of 26 IR, 20 had excellent response, 3 remained in the IR group, and 3 developed an incomplete response. In the IRTgAb+ group (*n* = 187), 45 achieved an excellent response, 141 showed IR, and 1 had an incomplete response after two years. Among 141 IR, 88 achieved an excellent response, 51 showed IR, and 2 had an incomplete response after five years. In the IRImaging+ group (*n* = 25), 19 patients had an excellent response and 6 remained IR after two years, and among these 6 cases, 3 showed excellent responses, 2 were IR, and 1 had an incomplete response in the fifth year after therapy (Figure 2).

### 3.2. Features Associated with Excellent Response 2 and 5 Years after Therapy

Clinical and pathological variables were investigated to study a possible association with developing an excellent response from IR. In the univariate analysis, only 1-year stimulated Tg (sTg) was significantly associated with excellent response status two years after therapy (*p* < 0.001). No other clinical or epidemiological features were related to disease status. sTg was also confirmed to be significantly correlated in the multivariate analysis (*p* < 0.001). On the other hand, if we analyze the 5-year status of the disease, in the univariate analysis, sTg and nodal disease at diagnosis were correlated with excellent responses. In the multivariate analysis, sTg and nodal disease were confirmed to be independent variables (Table 2). No other variables were associated with final disease status.

### 3.3. Survival Outcomes

At a median follow-up period of 84 months, relapse or progression of the disease occurred in 16 patients with an average time of 55 months (range: 11–111 months). Relapse/progression happened more frequently in patients with IRTg+ compared to IRTgAb+ and IRImaging+ subgroups (*p* = 0.001). In fact, nine patients with IRTg+ developed relapse/progression during the course of the disease, while only six with IRTgAb+ and one with IRImaging+ developed the same outcome. The median PFS was 83 months (range 11–149 months); 2-year and 5-year PFS were 99.6% and 96%, respectively. Considering the three IR subgroups, PFS was significantly shorter in the IRTg+ group compared to the IRAbTg+ and IRImaging+ groups. Two-year PFS was 99% in the IRTg+ group, 100% in IRTgAb+, and 100% in IRImaging+ (*p* = 0.003); 5-year PFS was 90% in IRTg+, 99.3% in IRTgAb+, and 95% in IRImaging+ (*p* < 0.001 and *p* = 0.001, respectively).

Upon applying the Receiving Operator Curve (ROC) analysis (Figure 3), we derived the best cut-off values for 1-year sTg of 3.3 ng/mL to predict the risk of relapse/progression.

Four patients with sTg < 3.3 ng/mL developed relapse/progression, while 5 had sTg > 3.3 ng/mL (*p* < 0.002). PFS in patients with sTg < 3.3 ng/mL was not significantly different from IRTgAb+ and IRImaging+ patients (Figure 4).

## 4. Discussion

Our study investigated the natural history and clinical outcomes of DTC patients with an IR one year after initial therapy (surgery plus radioiodine). IR is an intermediate response that falls between an excellent response and an incomplete response, and it needs personalized follow-up and investigations to detect the evidence of structural disease. In our population, IRs were recorded in approximately 13% of patients studied in the period of 2010–2017, a smaller period compared to the literature findings [3]. In the literature, the risk of structural disease is reported at approximately 15–20% [3], while our study showed a lower risk (approximately 4%). This is likely due to the shorter follow-up time of 7 years that we included in our study and the high percentage of low-risk disease patients included. DTC is a disease with optimal outcomes and a low risk of progression, and for this reason, a longer follow-up time is desirable. Moreover, approximately 20% of our patients remained as IRs at the last routine follow-up visit, not excluding the possibility that some incomplete responses were missing.

Recent evidence suggests that disease status after the initial therapy is more accurate in predicting recurrence compared to post-surgery evaluation based on pathological findings [4]. The response to the therapy re-staging system provides a more accurate, real-time individualized risk estimate, which can be used to guide the appropriate frequency and type of surveillance assessments [8,9].

We observed that most patients evolved an excellent response over time or remained IR, without any further treatment. This evidence is in agreement with other studies [10,11,12]. Consequently, the empirical use of RAI therapy in these cases is not supported by strong evidence and seems excessive, also considering the potential early and late side effects of RAI. It would be ideal to reserve further RAI treatments only in selected patients, and the identification of these patients is a key point. Of course, all patients in this study received RAI therapy, which may influence the results.

IR, by definition, includes patients with different features related to biochemical and radiological findings: IRTg+, IRTgAb+, and IRImaging+. These subgroups are widely different despite belonging to the same class of response. We demonstrated that structurally persistent/recurrent disease was most commonly present in the Tg+ group (10%) rather than in the other two groups. Only six patients in the TgAb+ group and one patient in the Imaging+ group had structural persistent/recurrent disease. This highlights the need for a more individualized follow-up strategy in the management of patients with an IR, especially in Tg+ patients. Moreover, among them, a Tg cutoff value of 3.3 ng/mL may help in discriminating those with worse survival. In our analysis, this value helped to discriminate the aggressiveness of disease and to predict the outcome.

A previous study derived a similar Tg threshold of 3.1 ng/mL [13], analyzing a more numerous sample of IRTg+ cases and a longer follow-up time period, but a more heterogeneous population, including high-risk and metastatic DTC patients, which could influence the analyses. Malandrino et al. [14] did not find an absolute value of Tg as an outcome predictor, but they demonstrated that an increase in basal Tg > 40% correlated to a higher risk of progression of the disease. In our analysis, basal Tg had no role in predicting the final status of the disease, in contrast to sTg. Jeong et al. [15] demonstrated that pre-RAI sTg levels were the only risk factor for developing IR after RAI, but they only studied a limited number of IR cases (*n* = 64) and did not follow up over a prolonged period of time. A recent study showed that sTg was superior to basal Tg in the evaluation of the treatment response after RAI therapy in low- and intermediate-risk patients [16]. However, more solid analyses are needed to confirm or controvert this evidence.

Another factor affecting the disease status five years after therapy is the presence of nodal disease at diagnosis. This evidence was previously suggested by other studies. [14,17]. A high nodal involvement (more than three lymph nodes) presented an elevated frequency of structural or biochemical disease in patients with an initial indeterminate response to therapy with RAI [17]. Malandrino et al. [14] associated the presence of >5 nodes with a higher risk of structural disease.

It is important to underline that, in this study, most patients recruited had a low-risk disease (T1–T2) for the definition of a group of patients with optimal prognosis and low risk of recurrence. This selection could influence the Tg threshold and has to be considered in future studies.

Thus, we may suggest that more frequent follow-ups via blood samples (Tg, TgAb) and imaging examinations (i.e., ultrasound) may be crucial in patients with IRTg+, especially with sTg greater than 3.3 ng/mL, whereas a less strict follow-up management strategy may be adopted for IRTgAb+ and IRImage+ groups.

The potential usefulness of a new RAI treatment is difficult to evaluate considering the retrospective nature of the study and the different guidelines in use at the time of patient enrollment. This topic should be studied in subsequent multicentric and/or prospective analyses.

Our study had some limitations, such as the retrospective nature of the research, the relatively short follow-up time period considering the course of DTC disease, and the low rate of adverse events, which are used to perform ROC curve analyses.

Moreover, nowadays, rh-TSH stimulation is indicated in the study of patients with low- to intermediate-risk disease and it is well demonstrated that sTg, in this case, is relatively lower than sTg after withdrawal. Thus, the sTg derived in our study needs to be carefully considered in the case of patients undergoing rh-TSH stimulation. We believe that different specific Tg cut-off values after rh-TSH injection would be more appropriate. Moreover, we can exclude the potential for the treatment of these patients before the introduction of rh-TSH.

## 5. Conclusions

Patients with DTC and IR 1 year after initial therapy frequently achieved an excellent response during the course of the disease; stimulated Tg may help to predict the latest disease status. The risk of progression is more frequent in Tg-positive patients with Tg values greater than 3.3 ng/mL.

## Figures and Tables

**Figure 1 cancers-15-01270-f001:**
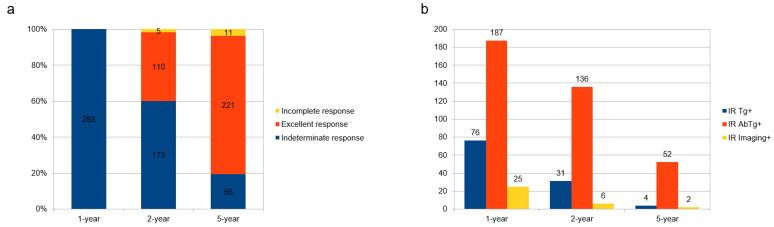
Distribution of treatment response (**a**) and distribution of indeterminate response (IR) subgroups over time (**b**).

**Figure 2 cancers-15-01270-f002:**
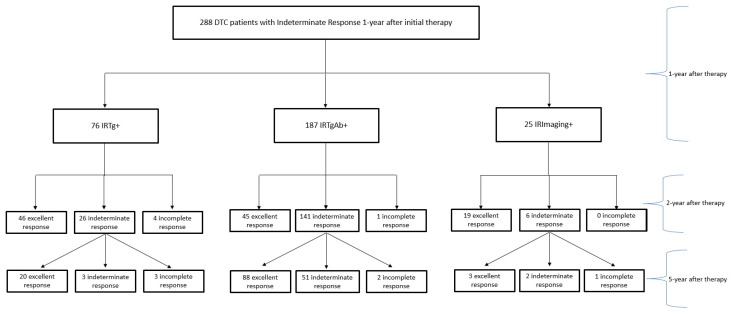
Flowchart of disease evolution over the years.

**Figure 3 cancers-15-01270-f003:**
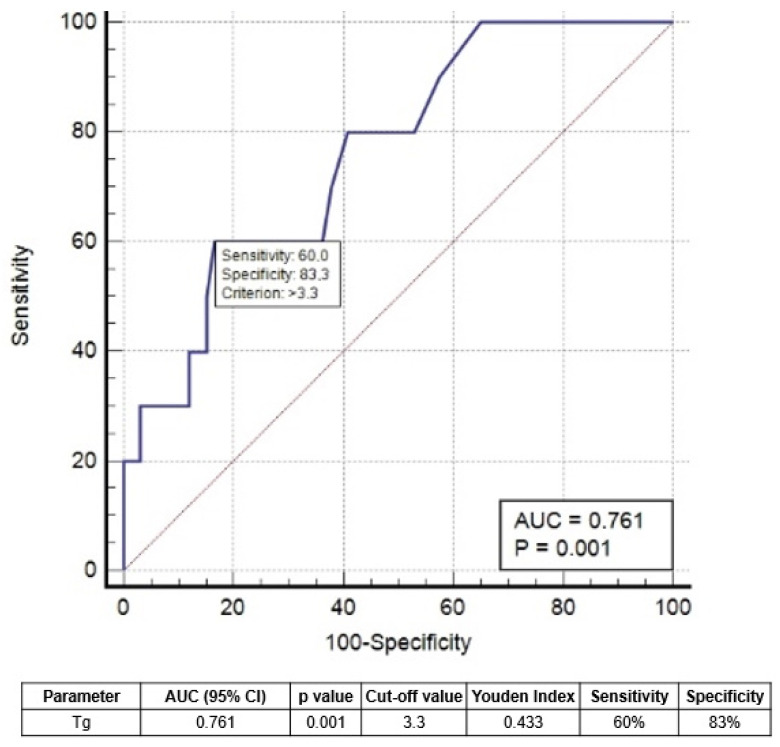
ROC curve analysis to determine the best threshold of sTg to predict progression/relapse.

**Figure 4 cancers-15-01270-f004:**
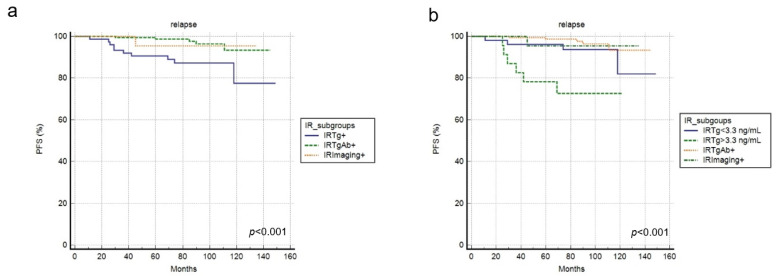
PFS curves according to IR subgroups and sTg level. (**a**) Kaplan Meier curve analyzing the three IR subgroups. (**b**) Kaplan Meier curve analyzing the four IR subgroups.

**Table 1 cancers-15-01270-t001:** Baseline features of patients with indeterminate responses after initial therapy (*n* = 288).

	Mean ± SD (Range)	Patients, *n* (%)
Age	49 ± 15 (18–84)	
Gender		
Female		235 (82%)
Male		53 (18%)
Histotype		
Papillary		118 (41%)
Follicular variant of Papillary		92 (32%)
Follicular		53 (18%)
Aggressive variant		25 (9%)
Tumor size, mm	16 ± 13 (1–130)	
Multicentricity		168 (58%)
Central lymphadenectomy		100 (35%)
Lateral lymphadecentomy		59 (20%)
TNM		
Tx		2 (1%)
T1b		214 (74%)
T2		50 (17%)
T3		22 (8%)
N0		226 (78%)
N1a		42 (15%)
N1b		20 (7%)
ATA initial risk		
low		217 (75%)
intermediate		71 (25%)
Tg at the time of ablation, ng/mL	3.4 ± 5.5 (0.2–114)	
TgAb positivity at ablation		198 (69%)
First RAI activities administrated, GBq	2.85 ± 1.1 (0.9–4)	

SD: Standard deviation; RAI: Radioiodine; GBq: GigaBequerel; ATA: American Thyroid Association; Tg: Thyroglobulin; Ab: Antibodies.

**Table 2 cancers-15-01270-t002:** Univariate and multivariate analysis of factors associated with final disease status.

2-Year Treatment Response					
Variable			Univariate Analysis	Multivariate Analysis	
	ER (*n* = 110)	Not ER (*n* = 178)	*p*-value	HR (95%CI)	*p*-value
Gender F:M	86:24	149:29	0.241		
Age at diagnosis	48.2	48.7	0.807		
Tumor size	15.4	17	0.333		
Multicentricity	60	108	0.768		
ATA initial risk low	85	132	0.198		
T stage: T1–2 vs. T3	102:8	162:16	0.501		
N metastases	20	42	0.470		
Tg at ablation	6.7	5.6	0.324		
First RAI administrated, GBq	2.5	2.7	0.191		
1-year sTg	1.8	2.7	<0.001	5.02 (2.454–8.909)	<0.001
**5-Year Treatment Response**					
**Variable**			**Univariate Analysis**	**Multivariate Analysis**	
	ER (*n* = 221)	Not ER (*n* = 67)	*p*-value	HR (95%CI)	*p*-value
Gender F:M	175:46	60:7	0.106		
Age at diagnosis	49	47	0.320		
Tumor size	15.8	18.1	0.236		
Multicentricity	132	36	0.200		
ATA initial risk low	172	42	0.509		
T stage: T1–2 vs. T3	204:17	60:7	0.765		
N metastases	38	24	0.001	2.345 (1.134–3.190)	0.012
Tg at ablation, ng/mL	6.8	7.9	0.599		
First RAI administrated, GBq	2.8	3.1	0.065		
1-year sTg, ng/mL	1.9	3.3	0.002	4.102 (2.601–7.879)	0.003

ER: Excellent response; F: Female; M: Male; Tg: Thyroglobulin; RAI: Radioiodine; sTg: Stimulated thyroglobulin; ATA: American Thyroid Association; HR: Hazard ratio.

## Data Availability

The data can be shared upon request.

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
