# Peer review of "Temporal Evolution and Prognostic Role of Indeterminate Response Sub-Groups in Patients with Differentiated Thyroid Cancer after Initial Therapy with Radioiodine"

_cancers, 2023, doi:10.3390/cancers15041270_

Round 1
Reviewer 1 Report
The authors present 5 year follow up data on 288 patients with intermediate response to initial treatment of DTC, stratified into three groups based on the reason for the intermediate response. The results may allow clinicians to further individualise follow up.
Overall the study is of interest to help guide the management of these patients, although there are a few points that would benefit from clarification.
1. I am not clear what the relevance of the 2176 patients who received RAI is given the rest of the paper only focuses on the 288 with IR.
2. The histology is given including 25 with more aggressive histological subtypes, but there is no sub-analysis according to histological risk and low vs high risk histology was not included in the analysis - this would have been of interest. At least should stage how many of the recurrences were in those with high risk histology.
3. All patients were treated with thyroxine withdrawal rather than recombinant TSH, although this would not now be standard practice in many centres for low risk disease. This should be discussed as a limitation.
4. In the discussion the authors suggest more limited RAI may be an option. However as all participants received RAI (and see point 3) it is hard to see how this conclusion can be reached and should be removed.
Author Response
The authors present 5 year follow up data on 288 patients with intermediate response to initial treatment of DTC, stratified into three groups based on the reason for the intermediate response. The results may allow clinicians to further individualise follow up.
Overall the study is of interest to help guide the management of these patients, although there are a few points that would benefit from clarification.
- Thanks you very for your comments and observations. I have modified the manuscript following them
1. I am not clear what the relevance of the 2176 patients who received RAI is given the rest of the paper only focuses on the 288 with IR.
- This number was put only to clarify how patients were treated in the period included in the study and so derive the rate of IR patients. This is a descriptive data.
2. The histology is given including 25 with more aggressive histological subtypes, but there is no sub-analysis according to histological risk and low vs high risk histology was not included in the analysis - this would have been of interest. At least should stage how many of the recurrences were in those with high risk histology.
- I really understand your point. First, I have to say that 25 cases of 288 is a very low number with weak statistical analysis
3. All patients were treated with thyroxine withdrawal rather than recombinant TSH, although this would not now be standard practice in many centres for low risk disease. This should be discussed as a limitation.
- Yes, this is true. In our hospital, rh-TSH was introduced in 2015. While the patients of this study are included between 2010 and 2017…However, I have added a few lines in the discussion about this point
4. In the discussion the authors suggest more limited RAI may be an option. However as all participants received RAI (and see point 3) it is hard to see how this conclusion can be reached and should be removed.
- I understand your point. Our comment was a personal opinion derived by the results of the study. However, I have changed a little the discussion.
Reviewer 2 Report
In this very interesting manuscript, authors are describing the clinical outcome of 288 patients with differentiated thyroid carcinoma and indeterminate response after thyroidectomy and ablative radioiodine therapy. The manuscript is well structured and well written. The knowledge of patients´ outcome, especially the fact that many patients spontaneously achieve an excellent response, is very helpful for both treating physicians as well as for patients.
Of course, the study has various limitation, the most important one being a retrospective analysis.
1. Unstimulated Tg or stimulated Tg can be used to evaluate therapy response. It is important to note if stimulated Tg was additionally performed 2 or 5 years after ablation as it may impaxt the axxurac of response classification. If it was not performed it should be mentioned as a limitation.
2. The decision to perform an additional treatment for the 54 patients who received the new RAI treatment was based on what criteria
3. Relapse or progression of disease was observed after median follow-up period of 84 months (7 years). However, the follow-up time is stated as 5 years in the discussion (page 9, l. 225). Could relapses or progression have been overlooked?
4. The methods mention that all patients received TSH suppression during follow-up. Do you believe that all patients with an indeterminate response benefit from TSH suppression? At what point should the TSH level be decreased to the lower normal range?
5. Were patients with unifocal pT1a included in the study? What criteria were used to decide on radioablation treatment for these patients?
6. Additional imaging can be helpful to identify lymph node (or even distant metastases). The number of cases in which MRI or PET/CT was performed is not specified in the information given. The effectiveness of MRI or PET/CT in identifying (lymph) node metastases is also not specified in the information provided.
7. The information regarding the relationship between indeterminate response and the dose of iodine given (1,0 Gbq vs 3,7 GBq) is not specified in the information provided. Did you see correlation?
8. Were patients reoperated in case of nonspecific findings in imaging or elevated thyreoglobluin?
Author Response
In this very interesting manuscript, authors are describing the clinical outcome of 288 patients with differentiated thyroid carcinoma and indeterminate response after thyroidectomy and ablative radioiodine therapy. The manuscript is well structured and well written. The knowledge of patients´ outcome, especially the fact that many patients spontaneously achieve an excellent response, is very helpful for both treating physicians as well as for patients.
Of course, the study has various limitation, the most important one being a retrospective analysis.
- Thanks you very for your comments and observations. I have modified the manuscript following them
1.Unstimulated Tg or stimulated Tg can be used to evaluate therapy response. It is important to note if stimulated Tg was additionally performed 2 or 5 years after ablation as it may impaxt the axxurac of response classification. If it was not performed it should be mentioned as a limitation.
- I understand your point. This is true sTg was not performed at 2 and 5 year in all cases. I have added this point in the discussion. However, for the definition of classes also not sTg can be used.
2The decision to perform an additional treatment for the 54 patients who received the new RAI treatment was based on what criteria
- Yes, it is true, I have added few lines explaining the reasons. “The decision to perform a new therapy was done by the nuclear medicine physician according to imaging and biochemical evidences.”
3.Relapse or progression of disease was observed after median follow-up period of 84 months (7 years). However, the follow-up time is stated as 5 years in the discussion (page 9, l. 225). Could relapses or progression have been overlooked?
- Sorry, this was a typo. I have corrected it.
4The methods mention that all patients received TSH suppression during follow-up. Do you believe that all patients with an indeterminate response benefit from TSH suppression? At what point should the TSH level be decreased to the lower normal range?
- Yes, in the period included (2010-2017) the indications of our center were to have TSH suppressed. The last ATA guidelines changed this point, but at that period were not available.
5.Were patients with unifocal pT1a included in the study? What criteria were used to decide on radioablation treatment for these patients?
- No, patients with T1a was not included in the study because they didn’t received RAI as guidelines said.
6Additional imaging can be helpful to identify lymph node (or even distant metastases). The number of cases in which MRI or PET/CT was performed is not specified in the information given. The effectiveness of MRI or PET/CT in identifying (lymph) node metastases is also not specified in the information provided.
- This point is not the core of the study and this is the reason why we don’t include these data. Moreover, only few cases performed MRI or PET/CT
7.The information regarding the relationship between indeterminate response and the dose of iodine given (1,0 Gbq vs 3,7 GBq) is not specified in the information provided. Did you see correlation?
- Good observation. I have added this analysis.
8.Were patients reoperated in case of nonspecific findings in imaging or elevated thyreoglobluin?
- Fifty-four patients received a new RAI treatment, and belonged to the following groups: 24 with IRTg+, 27 IRTgAb+ and 3 IRImaging+. The decision to perform a new therapy was done by the nuclear medicin physician according to imaging and biochemical evidences.
Reviewer 3 Report
This manuscript analyzes long-term outcomes of patients treated with surgery and radioactive iodine for differentiated thyroid cancer and classified as having an indeterminate response to therapy at 1 year of follow-up. This retrospective analysis is well-designed and the findings are plausible and meaningful. Patients were followed for at least 60 months, which is excellent.
I have some minor comments/questions for the authors.
1. Patients with a high risk of structural disease recurrence were not included in the study. Please explain why this is the case. Is it plausible that none of the high-risk patients had an indeterminate response to therapy?
2. The stimulated sTg cutoff of 3.3 ng/ml was derived from the very small number of adverse events, which is a limitation. Please acknowledge this. Also, many practices do not use stimulated Tg routinely for follow-up (especially using thyroid hormone withdrawal), which limits the utility of this finding.
3. The authors mentioned that “In our analysis, basal Tg had no role in predicting the final status of disease, contrary to sTg”, but no data on basal Tg is presented in the Results section.
4. Line 70. Non-specific imaging findings are not part of the definition of the 2) IRTgAb+ group
Author Response
This manuscript analyzes long-term outcomes of patients treated with surgery and radioactive iodine for differentiated thyroid cancer and classified as having an indeterminate response to therapy at 1 year of follow-up. This retrospective analysis is well-designed and the findings are plausible and meaningful. Patients were followed for at least 60 months, which is excellent.
I have some minor comments/questions for the authors.
- Thanks you very for your comments and observations. I have modified the manuscript following them
1Patients with a high risk of structural disease recurrence were not included in the study. Please explain why this is the case. Is it plausible that none of the high-risk patients had an indeterminate response to therapy?
- In the study design, we decided to include patients with indeterminate disease 1 year after first therapy and as you anticipated in the question high-risk patients usually had a more negative response after 1-year. Really, in my center some cases are present but I prefer to not include them in this study because high risk patients for definition are a diffent class or risk. However, I stressed this point in the discussion.
2.The stimulated sTg cutoff of 3.3 ng/ml was derived from the very small number of adverse events, which is a limitation. Please acknowledge this. Also, many practices do not use stimulated Tg routinely for follow-up (especially using thyroid hormone withdrawal), which limits the utility of this finding.
- Yes, you are right, the number of events was not high (like in previous works). This is a limitation. Also the absence of rh-TSH stimulation data my reduce the impact of this work. However, I believe that the results are original and of clinical interest. In our hospital, rh-TSH was introduced in 2015. While the patients of this study are included between 2010 and 2017….
However, I have some lines in the limitations section of discussion.
3.The authors mentioned that “In our analysis, basal Tg had no role in predicting the final status of disease, contrary to sTg”, but no data on basal Tg is presented in the Results section.
- Sorry, I’m not sure to have understood this question. In Table 2 I have put also Tg at ablation...
4.Line 70. Non-specific imaging findings are not part of the definition of the 2) IRTgAb+ group
-OK I have deleted it
Reviewer 4 Report
This is a very good article, well written and illustrated. Although the study has important limitations, recognized by the authors, its data are convincing and the conclusions adequate.
Some observations may help mprove the quality of the publication.
1. The fact that most patients are at low risk, including cases of millimetric tumors, deserves further discussion. Would microcarcinomas present a different evolution and therefore different management? I suggest a few word addressing this issue in the Discussion.
2. Would a trend toward an increase or decrease in serum TG help categorize patients?
3. It is noteworthy that a service that provides access to sophisticated exams still removes medication from patients to stimulate TG elevation. And they still use T3. Why did the authors not use recombinant TSH?
4. There is no mention of approval of the study by any Ethics Committee or agreement of patients after signing the Free and Informed Consent Form.
Author Response
This is a very good article, well written and illustrated. Although the study has important limitations, recognized by the authors, its data are convincing and the conclusions adequate.
- Thanks you very for your comments and observations. I have modified the manuscript following them
Some observations may help mprove the quality of the publication.
1. The fact that most patients are at low risk, including cases of millimetric tumors, deserves further discussion. Would microcarcinomas present a different evolution and therefore different management? I suggest a few word addressing this issue in the Discussion.
- Yes, it is true, the number of low-risk patients is very high and this “selection” may influence the results. I have added something about this point in the discussion section.
2. Would a trend toward an increase or decrease in serum TG help categorize patients?
- Good question!!!! This parameter was not evaluated in this study because it would be very difficult to calculate. Moreover, only in Tg-group this parameter could have a role. Instead, the aim of this study was to analyze and compared the different indeterminate response groups.
3. It is noteworthy that a service that provides access to sophisticated exams still removes medication from patients to stimulate TG elevation. And they still use T3. Why did the authors not use recombinant TSH?
Thanks a lot for this question. In our hospital, rh-TSH was introduced in 2015. While the patients of this study are included between 2010 and 2017, so many of them didn’t have the opportunity to perform rh-TSH stimulation. Moreover, I believe that for a high reproducible evidence is important the all patients had similar analyses.
4. There is no mention of approval of the study by any Ethics Committee or agreement of patients after signing the Free and Informed Consent Form.
- In the last section of the manuscript I’ve reported these data “Informed Consent Statement: Informed consent was obtained from all subjects involved in the study. Data Availability Statement: the data can be shared up on request”
Round 2
Reviewer 2 Report
Thank you for the revised manuscript. However, some comments are still insufficiently answered. Please be more precise. English of your answers have to be improved.
1. Unstimulated Tg or stimulated Tg can be used to evaluate therapy response. It is important to note if stimulated Tg was additionally performed 2 or 5 years after ablation as it may impact the accuracy of response classification. If it was not performed it should be mentioned as a limitation.
“ I understand your point. This is true sTg was not performed at 2 and 5 year in all cases. I have added this point in the discussion. However, for the definition of classes also not sTg can be used.”
Page 4, ll. 139 ff: In not all cases, patients performed WBS and sTg at 2-year prefering not stimulated Tg; while at 5-year as institutional protocol, all patients performed a WBS and sTg. Neck ultrasound and/or other imaging studies according to patients features wer done at 2 and 5 year.
Unfortunately, your English is not understandable. Did you or did you not perform a WBS and stimulated Tg after five years? If it was performed only in a subgroup, the number of patients should be mentioned.
2. The decision to perform an additional treatment for the 54 patients who received the new RAI treatment was based on what criteria
Yes, it is true, I have added few lines explaining the reasons. “The decision to perform a new therapy was done by the nuclear medicine physician according to imaging and biochemical evidences.”
Well, it is clear that the decision was made by a physician. However, you included 288 patients and re-therapy was only performed in 54 patients. What were the criteria in these 54 patients? Could you please specifiy the imaging and biochemical evidences.
4 The methods mention that all patients received TSH suppression during follow-up. Do you believe that all patients with an indeterminate response benefit from TSH suppression? At what point should the TSH level be decreased to the lower normal range?
- Yes, in the period included (2010-2017) the indications of our center were to have TSH suppressed. The last ATA guidelines changed this point, but at that period were not available.
Please state in the limitation that patients received TSH suppression and were overtreated in comparison to nowadays standard.
7. The information regarding the relationship between indeterminate response and the dose of iodine given (1,0 Gbq vs 3,7 GBq) is not specified in the information provided. Did you see a correlation?
- Good observation. I have added this analysis.
RAI activity at ablation was not related with the risk to develop IR 1-year after the first therapy (2.85 Gbq vs 2.6 in whole population, p =0.345)
Did you test 2.85 versus 2.6 GBq I-131? This is redundant. Please test 1.1 GBq I-131 versus 1.85.-3.7 GBq. Did you notice any differences in these groups? Was diagnostic WBS more often positive in the group treated with lower radioactivity?
In this context, low-risk DTC and intermediate-risk DTC on page 2, ll 95 ff should be defined. In the discussion low-risk is defined as being T1-T2 (Page 8, ll. 279ff). However, the definition should be included in the methods instead of in the section discussion.
8. Were patients reoperated in case of nonspecific findings in imaging or elevated thyreoglobluin?
- Fifty-four patients received a new RAI treatment, and belonged to the following groups: 24 with IRTg+, 27 IRTgAb+ and 3 IRImaging+. The decision to perform a new therapy was done by the nuclear medicin physician according to imaging and biochemical evidences.
Could you please reply to my question? None of your patients were reoperated, correct?
Page 2, ll. 91 ff: Please replace by: Patients with unifocal microcarcinoma were excluded. Alternatively, this information can be added in the defition of low-risk patients (see above)
Please check the manuscript by a native speaker and correct various spelling and grammatical mistakes:
P4, l 133 imaging
P4, ,l 139 preferring
P4. L 141 were
P4, l 1652.85 GBq vs. 2.6 GBq
P9, l 294 stimulation
P9, l 294 diffuse in the study? What do you mean? Used in patients with
P9, l 297 carefully
P9, l. 297 can think? We believe?
P 9, l 298 suc-off? What do you mean? Do you mean Cut-off?
Author Response
1. Sorry for the errors. I have written again this part: “ In not all cases, patients performed WBS and sTg at 2-year preferring at this time not stimulated Tg; while 5 year after initial treatment, all patients performed a WBS and sTg as part of our institutional protocol. Neck ultrasound and/or other imaging studies according to patients features were done at 2 and 5 year.”
2. I understand your point. Here the explanations: “24 patients performed a new therapy due to the presence of positive sTg (ofter near to 10 ng/mL, always high then 5 ng/mL); 27 for the presence of positive TgAb especially when the trend of Tgab was stable and concomitant with not negative WBS; 3 for the presence of markedly positive WBS, like big thyroid remnants “
4 I undersatnd this point and I have changed the final. However, this a issue common of all patients treated before the introduction of rh-TSH…..
7 We performed these analyses (2,85 vs 2,6) because our aim to focus upon IR patients. So we compared the activity received by IR vs other patients. Of course we can make the analysed you asked despite it wasn’t the core of our study to investigate the “efficacy” or different I131 activities.
For the definition of class risk, I have added a phrase in the methods and the corresponding reference (ATA guidelines)
8 Yes I confirmed . No new operations
Moreover, I have corrected the grammatical errors that you noticed.